# How Have District-Based House Price Earnings Ratios Evolved in England and Wales?

David Paul Gray

Department of Accountancy Finance and Economics, University of Lincoln, Lincoln LN6 7TS, UK;
dgray@lincoln.ac.uk

**Abstract:** The central aim of this paper is to provide a baseline framework for describing the evolution of an affordability indicator at a district level, before and after the financial crisis of 2008. From the mid-1990s to 2019 house price-earnings ratio for England and Wales appear to have ratcheted-up, with the growth rate more rapid just before and a temporary decline just after the crisis. This masks a significant variation in evolutionary profiles. Following Turok and Mykhnenko in 2007 who set about exploring population trends in European cities, districts are classified into groups. Matching each district against ten stylised profiles, rather than cycles, persistent trends and single turning point paths in ratios are the norm. An asset-price model projects that finance will favour those bright futures so that spatial-sorting of those with high human capital leads to some districts benefitting from lending criteria out of line with others.

**Keywords:** house price-earnings ratios; local authority districts; Kendall's *W*; Jonckheere-Terpstra test; an asset-price model

## 1. Introduction

Both the Bank of England and Central Bank of Ireland regulate the disbursement of mortgages through ratios (Jamei and O'Brien 2017). One can see two dimensions to this: first, to reduce the risk of some parts of the country dislocating from normal 'affordability' measures; second, if affordability ratios rise above their long-term averages, it could herald a crisis. Both countries saw a house price bubble in the mid-2000s. The Bank of Bank of England (2015) believes that peaks in house price-disposable earnings ratio (HPER) precede a financial crisis by 1–2 years.

Spatial co-movements in HPERs have been explored, but not to the same extent as house prices. A gap in our knowledge is what happened to this measure of affordability at the local level in the UK during and after the 2004–2008 bubble bursting. There are two themes in the literature which are considered. Gregoriou et al. (2014) see a dislocation between income and price. Gray (2022) finds both a general rise in the house price-earnings ratios and also a steepening of the 'gradient' of ratios over 2004 to 2019 at the district level. How this steepening evolved and is distributed across space are gaps in our knowledge. Pitros and Arayici (2017) report upswings and downswings in the ratio in the UK at the regional level. Is there evidence of these swings at the local level, and if so, where?

Turok and Mykhnenko (2007) (hereon T&M) set about exploring 'Resurgent Cities' in Europe, where they discerned nine long-term paths or trends in city populations over 45 years. This set was applied specifically to UK cities by Gray (2021) over 1981–2018, using Centre for Cities' definitions of British cities as primary urban areas (PUAs),[1] and exploring relative population change. What is found is that UK cities commonly have trends in rising or falling prominence, which can be explained using path-dependency theory (Martin et al. 2014). Long-term growth of a centre or region would influence future population movements, relative house prices and local affordability. This points to persistent trends in affordability that are peculiar to cities.

Following T&M's approach, the central aim of this paper is to provide a baseline framework of the house price-earnings ratio paths. Given population trajectories identifying whether there are distinct HPER paths, such as those typified by T&M's approach, particularly with a spatial clustering, should help underpin policy initiatives around the perception of affordability. Whether these trends are affected by the 2004–2008 period is explored.

The paper is structured as follows. First, there is a review of affordability measures in the run up to, and after, the 2008 period. This is followed by theoretical contributions to the house price-income nexus. The methods section covers the adaptions of non-parametric tests. The Jonckheere-Terpstra (hereon J-T) test for ordered alternatives is applied to growth in district HPERs grouped into three regional sets, distinguishing between City and non-City districts. A second version of this explores the HPER levels of 10 posited T&M paths at certain time points. The district set are allocated to a path based on Euclidean distances. Pairwise Kendall's *W* statistics are used to test for stability in the HPERs' hierarchy. This is followed by a discussion of the district data. Results and conclusions follow.

## 2. Recent History of Regulation and Affordability

For many developed countries, financial deregulation over 1980–2007 strongly magnified the impact of the financial sector on the occurrence of booms, which was fortified by international liquidity (Agnello and Schuknecht 2011). Hay (2009) compares the UK and Irish experiences of house price inflation. He argues that for different institutional reasons, from the early 1990s, both pursued consumer-led growth strategies that fed on raised private debt levels. However, it was the incentives around allocating mortgage debt in both countries that inflated prices, propitiously. Addison-Smyth and McQuinn (2010) posit that in both the UK and Eire, the ability of credit institutions to access funding from abroad, post-2000, increased average mortgage levels, so that prices in 2008, on average, were 30% greater than what they would have been if the wholesale funds had not been available.

Measures of affordability relate house price to incomes. One international survey, Demographia International Housing Affordability Survey (2020), generated a 'median multiple'; the house price to the *household* income. For 2019, the values for the UK and Ireland for all markets were 4.5 and 4.1, respectively. Values between 4.1 and 5.0 were described as seriously unaffordable. The UK's Office for National Statistics (ONS) generates HPERs for each Local Authority District in England and Wales that are distinct from Demographia's (and the ones the Bank of England regulates). The ONS's are all actual prices, including those purchasing without a mortgage. The annual earnings cover all district residents' work-based earnings, including those not engaged in house purchase. The median value for 2019 for England and Wales was 7.7.

Researchers examining time series data on HPERs commonly use Nationwide Building Society's or Halifax Bank's house price data. For example, to generate HPERs, Pitros and Arayici (2017) and Gregoriou et al. (2014) use the earnings values of full-time males taken from the ONS. This approach to generating spatial HPERs is much like Demographia's and the ONS's.

Although, since 2014, the Bank of England regulates the provision of mortgages through debt-to-income ratio (DTI) and the loan-to-value (LTV), rather than HPER directly, DTI is likely to provide similar indications of financial distress as HPER. The ratio reported by UK Finance, the body representing big UK mortgage providers, and regulated by the Bank of England, is based on earnings of the borrower (or borrowers, where two incomes are used to assess the feasibility of repayment). UK Finance reported values for December 2019 for a first-time buyer (FTB) of a DTI of 3.54 and an LTV of 0.77, whilst for 1997, the values were 2.22, 0.88. More is lent per unit income whilst deposits have grown. The long-term correlation between the HPER and DTI[2] is around 0.96. For Others, they come to the market, usually with equity in the home to sell. Their borrowing per unit income has

risen from 1.99 to 3.3, whilst the LTVs have remained stable (0.65 to 0.68). The correlation between debt and HPER is lower at 0.89.

## 3. Prices and Theory

DiPasquale and Wheaton (1996, p. 44) analyse factors that affect land prices, which underpin dwelling prices. The first of these is the cost of capital. If it falls, this drives up all asset prices, including dwellings. Miles and Monro (2019) estimated that the sustained decline in real interest rates between 1985 and 2018 could account for all of the doubling of the house price-earnings ratio in the UK. There will be distributional variations in how finance affects prices. Himmelberg et al. (2005) argue that house prices are more sensitive to changes in real interest rates in rapidly growing cities. They posit a one percentage point decline in real interest rates could raise house prices by as much as 19% in a location that averaged 1.8% price growth, and 33% in a 3.8% price growth market.

An asset pricing model projects that a determinant of fundamental house value is the rental yield (rent ÷ price) (Bank of England 2015). The rental stream will be a function of the local level of productivity. Gal and Egeland (2018) aver that UK regional productivity disparities have been increasing since the early 2000s. They suggest the low productivity of UK regions is, to a large extent, driven by its major cities. Northern cities should have HPER evolutions that echo weak productivity paths. Martin et al. (2014), point to locational-sorting, the tendency for individuals to self-sort across space, so that highly productive, knowledge workers are attracted to certain cities, which further aligns with Fielding's (1992) escalator region. This is a particular draw for the young, highly-educated [graduate] migrant. Such regions provide the context within which residents achieve accelerated upward social mobility through movement within the region's labour and housing markets. These agents have a long-term view of expected returns from a location and so are less phased by current affordability levels (Swinney and Williams 2016), possibly because they benefit in the longer run (Fielding 1992).

DiPasquale and Wheaton also note that whether the current price is appropriate (the fundamental value) depends on expected rental income growth. With anticipated city productivity growth (Coulson et al. 2013; Van Nieuwerburgh and Weill 2010) or population growth (DiPasquale and Wheaton 1996; Glaeser and Gyourko 2005), rental yields are expected to have a greater locational value in the future. The current local house price will be higher (Sinai 2010) and so too will the HPER.

Martin et al. (2014, 2018) utilise an evolutionary perspective, where agglomeration economies trace out productivity development paths. Path-dependency results from a constellation of structures, institutions, and knowledge locking a region into a growth path. Accordingly, a disadvantageous industrial mix could constrain a city to persistent poor performance. De-industrialisation has provided an explanation of sluggish growth in the northern cities for some decades (Martin et al. 2018; Pike et al. 2016). Servillo and Russo (2017) find an 'embedded' nature of smaller settlements within urban and regional systems, which are likely to be path-dependent. The city performance could infect the broader region. Beatty and Fothergill (2020) find Britain's older industrial towns are in relative decline wherever they are located.

DiPasquale and Wheaton's third factor is risk. With a rise in price, those home-movers who are trading-up are in a stronger position in the market to buy their next property (Stein 1995). As equity is enhanced, there are lower bank agency costs. Larger advances would fortify the general price rise, apparently reducing the risk of greater lending, and acting as a financial accelerator (Aoki et al. 2004). By lending more to purchase the same properties over time, lenders inflate the wedge between prices and incomes, which is more likely to affect high house-priced areas. Greater lending for the same income could result from a perception of a less risky area or borrower. Spatial inequalities in HPERs could be viewed as reflecting risk-adjusted returns to a dwelling purchase (Gray 2022; Sinai 2010).

As house prices are more volatile than incomes, HPER are likely to follow the same cyclical patterns as prices. Pitros and Arayici (2017) find UK HPER cycles are similar to

price cycles: they are highly synchronised, with both having a complete cycle of 19 years on a peak-to-peak basis. Finding regional HPERs non-stationary, Gregoriou et al. (2014) aver that there is a dislocation between income and price in the UK. The period 1983–2009, fits the real interest/cheap finance thesis (Ganoulis and Giuliodori 2011; Miles and Monro 2019). The widening of the price-income differential is likely to be distributed unevenly.

Local adjustment to a positive productivity shock would be seen in rents. Higher wage would attract more migration. Along with the local income effect there would be a bidding up of rents through competition for dwellings. The extent of this movement defines the extent of the housing market area (HMA). DiPasquale and Wheaton (1996) see a collection of dwellings of distinct qualities as part of the same urban HMA if there is a tendency towards a stable *structure* of prices. If an area is underpriced, potential buyers will seek-out that market, forcing a future price adjustment. Thus, the spatial co-movement of house prices and house price earnings ratios are to be expected.

The local structure of prices is analysed by McCann (2013) using a bid-rent model, where high-income groups outbid others for larger dwellings at the edge of a housing market area drawn by the attractor of lower density space. Rather than solely pecuniary factors affecting utility, locational amenities also feature in the spatial equilibrium model (Roback 1982). The presence of high rents must be offering some compensating advantage, such as space, access to shops, or place of work.

Although the structure of prices is expected to be stable within local areas, the productivity differentials and risk preferences across areas should result in distinctive growth patterns. Gray (2022) found that there are strong similarities between the 2004 and 2019 rank orders of district HPERs. Bifurcating the period into pre- and post-2008, in the first period, HPERs were found to converge, which was later trumped by greater divergence over 2009–2018. Post-2008 has seen regions outside of the south of England maintaining stable spreads of ratio, whilst inside they have been broadening, suggesting some districts pulling away at the top end.

## 4. Method

Given the locational housing choices of the more affluent (McCann 2013), it is posited that city HPERs are generally *lower* than their hinterlands'. Given the productivity growth and the fact that migration favours the south east (Martin et al. 2018; Pike et al. 2016) ratios become greater as London is approached. With the locational preference for space of the rich commuter, districts are sub-divided into City ($C$) and Other ($O$) and into broad regions with increasing distance from London. One can consider, jointly, a north-midland-south divide where average HPERs are ordered as: $South_O$ > $South_C$ > $Midlands_O$ > $Midlands_C$ > $North_O$ > $North_C$. The Jonckheere-Terpstra test for ordered alternatives can be applied to this when working in ranks, whereby the null of equal HPER ratios has the alternative that HPER ratios are a given order. Assume at time $t$ there are $q = 1 \ldots Q = 6$ groups, comprising $n_q$ districts, with the total number of districts being $N$. The first J-T test entails the null that HPER levels are distributed randomly so that the median HPER of any one group will be no different to any other. The alternative is that mean-ranks are ordered $\overline{R}_{1t}^l \leq \overline{R}_{2t}^l \leq \ldots \leq \overline{R}_{Qt}^l$. The $q$th group's mean-rank of HPER in levels ($l$) is $\overline{R}_{qt}^l = \frac{1}{n_q} \sum_{q=1}^{n_q} R_{qt}^l$.

The application of the test is as follows. The procedure entails first calculating Mann-Whitney counts. The counts entails $U_{iq} = \sum_{h=1}^{n_i} \#\left(x_{hi}, q\right)$, where $\#(x_{hi}, q)$ is the number of times $x_{hi}$ is smaller than values in sample $q$ where $i < q$. $J$ is the number of these counts, $= \sum_{i<q}^{Q} U_{iq} = \sum_{i=1}^{Q-1} \sum_{q=i+1}^{Q} U_{iq}$. The Jonckheere-Terpstra statistic is given by Siegel and Castellan

(1988) as: $\dfrac{J - \frac{N^2 - \sum_{q=1}^{Q} n_q^2}{4}}{\sqrt{\frac{1}{72}\left[N^2(2N+3) - \sum_{q=1}^{Q} n_q^2 (2n_q+3)\right]}} \sim N(0, 1)$. The levels inequality is the basis for the growth test.

Sala-i-Martin (1996) describes beta-convergence in a cross-section of regional economies as when there is a negative relation between the growth rate of income per capita and the initial level of income per capita. When group means $\theta_q$ are ordered such that the mean income levels can be seen as $\theta_1^l \leq \theta_2^l \leq \ldots \leq \theta_Q^l$ convergence would be implied by $\theta_1^g \geq \theta_2^g \geq \ldots \geq \theta_Q^g$, i.e., the order in growth rates ($g$) of the incomes is inverse to that of the levels in the initial period.

The J-T growth test entails using the alternative that mean-ranks of the $Q$ groups of growth rates are in the order $\overline{R}_1^g \geq \overline{R}_2^g \geq \ldots \geq \overline{R}_Q^g$. Growth is defined as the ratio of HPER at period $t + p$ divided by HPER at period $t$. This produces three test outcomes. There is equality of growth: growth is independent of levels. If that null is rejected and the J-T statistic is negative, that order of growth rates is deemed inversely related to levels, and there is convergence in group HPERs. As SPSS generates a two-tailed version, the positive J-T statistic is taken to indicate the rising paths are associated with higher HPERs, initially, or divergence.

Following T&M, ten profiles are created. With the period divided into three five-year phases, where the trend in a phase is presumed to be either D(own) or U(p), a DDU profile corresponds with T&M's *recent resurgence*, having two thirds of the period in relative decline and then five years of resurgence. *Recent decline* is coded as UUD; DUU corresponds with *long-term resurgence*; and DDD, *long-term decline* is the obverse of the resurgence, UUU. There are four others. *Medium-term decline* rises for about half the period then declines ($\wedge$). *Medium-term resurgence* ($\vee$) is the obverse. Although T&M have 9 paths or profiles there are two versions of W, *Growth set-back*, to account for more than one turning point. Both divide the period into four chunks of around 4 years. This really captures cyclical dynamics.

Consistent with T&M's consideration of the *share* of the European population living in cities, the district rates are divided by those of England and Wales (E&W) for that year. To address tail effects, aligned ranks are taken, where the relative ratios for each of the sixteen years from the 338 districts are ranked. Aligned ranks have the property of maintaining the size-order, whilst standardising the size-differential.

To classify the 338 district paths, each is set against all 10 profiles, selecting the one with the minimum Euclidean distances of the *z*-transformed series value. As this procedure is a stage in hierarchical cluster analysis, this can be performed easily by SPSS.

With known paths and unequal starting points, potentially, one could infer Beta-convergence. For example, assuming only DDD and UUU paths existed, and those with the former trajectory had a significantly higher median HPER initially [in 2004], the gap in HPER would be narrower, implying convergence. If, instead, the median HPER for the UUU group is the higher one, initially, this implies divergence from the outset.

Beyond just narrowing the gap, a logical implication of beta-convergence is leapfrogging (Quah 1996). Leapfrogging implies that with persistent growth paths, DDD and UUU, the convergence process will eventually change their rank-order positions. Leapfrogging implies the paths of the two groups crossed during the 2004–2019 period, inverting the order by the terminal period. By implication, an inversion would entail those with a UUU profile to have a lower initial and higher terminal mean-rank than DDD districts. Quah (1996) is critical of the absence of analysis of leapfrogging due to persistent paths. In between UUU and DDD there are 8 profiles.

The J-T trend test's ordered alternative of mean-ranks *a priori* is UUU < UUD < DUU < $\wedge$ < W1 < W2 < $\vee$ < UDD < DDU < DDD. The first three have rising paths and the last three having declining trends for the majority of the time. In a sense, the middle four order is contestable. If the district relative profiles are distributed randomly, the median HPER of any of the 10 will be no different from any other in the initial period. If the J-T statistic is significant, there is order to the HPER levels that corresponds with the paths. If negative, those with declining [rising] paths initially have higher [lower] HPER levels, which corresponds with a convergence alternative.

The terminal mean-ranks could also provide a useful structure. If the HPER mean-ranks are ordered as $\overline{R}^l_{UUU,t+p} \geq \overline{R}^l_{UDD,t+p} \geq \ldots \geq \overline{R}^l_{DDD,t+p}$, rising [declining] paths were linked to higher [lower] HPER levels in 2019.

A fourth procedure entails an examination of rank stability. Boyle and McCarthy (1997) introduced the use of the trend in rank concordance to reveal intra-distributional mobility. They use two variations of Kendall's *W* (coefficient of concordance) to measure the level of 'agreement' in the orderings of incomes of *N* countries across *p* periods. A score of 1.0 represents perfect 'agreement' and zero represents no 'agreement' in order. $\overline{R}_i$ is the mean-rank of the HPERs of district *i* defined as $\frac{1}{p} \sum_{t=1}^{p} R_{it}$. With a pairwise version ($W_2$) the mean-rank uses two periods, *t* = 1 and *t* = *p*. When the *N* mean-ranks are not different over time, the statistic will be zero. Kendall's pairwise concordance statistic $K_W$ can be defined as $\frac{12 \sum_{i=1}^{N} \overline{R}_i^2 - 3N(N+1)^2}{N(N^2-1)} \sim \chi_1^2$. Here, $W_2$ provides a measure of rank stability compared with a base year. If there is widespread leapfrogging, there is rank instability and low concordance.

One could look for consistency in inference from $K_W$ and J-T tests of trend and growth. $K_W$ cannot distinguish between stability and divergence, where districts are pulling away at the top or left behind at the bottom. Rank-preservation combined with persistent trends presents a potential inconsistency. It could be that the trends do not herald significant changes in locational preferences: the converging trends could be persistent but shallow. Alternatively, the trends could point to divergence or a combination indicating reverting to some order by an error correction mechanism.

## 5. Data

Local Authority District house prices and earnings are supplied by the UK's Office for National Statistics (ONS). Annual estimates of gross earnings are based on the reference tax year and relate to employees on adult rates of pay who have been in the same job for more than a year. Both workplace-based and residential district earnings are available. Work-based earnings are used to reflect the likely commuting of workers across district boarders. House Price Statistics for Small Areas are supplied to the ONS by the Land Registry, which provides a comprehensive record of property transactions in the UK.

Annual median HPERs of all dwellings are available from 1997 for 10 regions. Excluding the Isles of Scilly, which have intermittent data, there are values for 338 English and Welsh districts from 2004 to 2019. This covers the run up to, and the recovery from, the financial crisis of 2007–2008. The definition of City-districts is taken from the Centre for Cities' website.[3] The Centre uses a definition of a city as a primary urban area (PUA), which is drawn from the work of CURDS.[4]

With around 52 million across the 10 regions, the average population is twice that of the smallest, the North East. Regions are commonly polycentric but the north of England features a relatively large number of cities. Large cities are divided into districts. A district has an average population of 150,000 but the largest, Birmingham, is around seven times that. The sub-divisions of a city provide more colour, but it can isolate pockets of extreme opulence or poverty. Moreover, smaller nodes are merged into a district. Bournemouth, Christchurch and Poole were merged in the last iteration of district restructuring, perhaps better reflecting the nature of the growing conurbation that is polycentric. Despite the logic of such a merger, this highlights how divisions alter. One must be mindful of the impact on aggregation. The averaged district HPERs across a region can be inflated relative to the regional value. Without weights, harmonic means and non-parametric statistics would be more appropriate than standard ones.

Traced in Figure 1, the ONS was quoting a HPER of 3.55 for 1997 for England and Wales. This rose three income points to 2004, adding a further 0.6 points to its 2007 peak. An Agnello and Schuknecht (2011) bust, where the fall in house price should reflect order of the rise, was not evident in the ratio. Rather, the HPERs plateaued. From 2004 to 2013 HPERs

seemed relatively stable at around 6.76 ± 0.4, but at the end of their study the ratio then increased by 1.6 ratio points. The evolution could be seen as cyclic but wrapped around a trend. The Government Office for the Regions' values ranged between the highest ratio region (always London from 2009) and the lowest (always North East) by 1.2 to 7.1 points. The North East did not keep up with the E&W average. Even among the northern regions the drop in the ratio from 2007 was no more than 0.5 points. For all, HPERs were ratcheted up in the boom period and have not really unwound since. For southern regions, this ratchet continued.

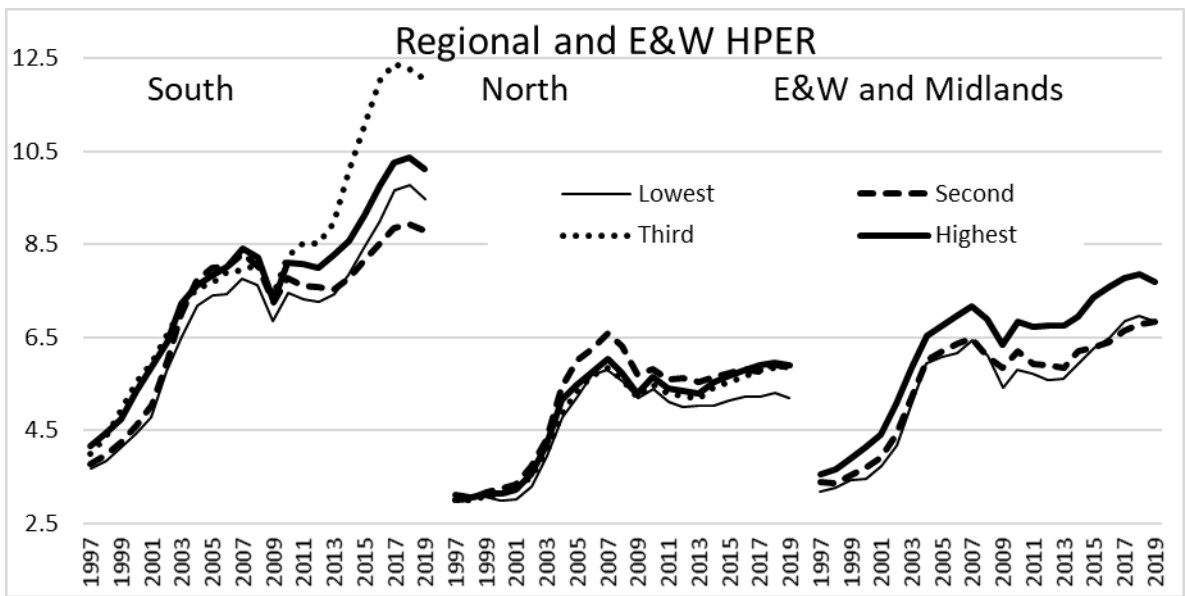

**Figure 1.** Regional House Price Earnings Ratio. South—Lowest: East of England < South West < London < Highest: South East; North—Lowest: North East < Wales < North West < Highest: Yorkshire-Humberside; E&W and Midlands—Lowest: East Midlands < West Midlands < Highest: E&W.

In 1997, the City districts' HPERs were lower than the Others (3.30 vs. 3.67). In 2004, the divide had grown (5.85 vs. 6.73), but, by 2019, the gap was smaller but maintained (7.50 vs. 8.05), which was consistent with the posited order.

## 6. Results

### 6.1. Kendall's Concordance

From Figure 2, $W_2$ for the regions indicated a very stable rank order. The one-year benchmarks every subsequent year's HPER against that of 1997. The 1997–1998 regional pairing had the $W_2$ value of 0.99; when matching 1997 with 2019 it was 0.98, having almost no change. This was repeated using 2019 as the base year (reverse 1 year), which provided much the same picture. Interestingly, although still a very high coefficient, agreement was lower around the peak of the house price bubble 2004–2008. In other words, the regional structure altered but reverted to something akin to 1997, a trough: the bubble period of 2004–2008 was unusual. All $W$-coefficients were significant at better than the 1% level.

There was a second rolling-$W_2$ set associated with a five-year interval. This would have a common value with a 1997–2001 pair. What was evident was that the regional set was very stable over a 5-year period.

The previous exercise entailed 10 regions. The district annual $W_2$ had the same order of value as the regional one. It declined, dropping to 0.92, again implying a very stable structure, but, with 34 times more territories, there was far more scope for shuffling. Repeating the process, but using 2019 as the base year, again revealed the mid-2000s were out of step with other periods.

Additionally, the district rolling $W_2$ revealed the same pattern as the regional set associated with a five-year interval. The structure of HPER appeared to move almost as one over multiple 5-year periods. Whether using regions or districts, the rank orders were, to a very high degree, in agreement. The difference between the regional and district one-year pairings would reflect leapfrogging within regions, but the 5-year series suggest this was of a low order.

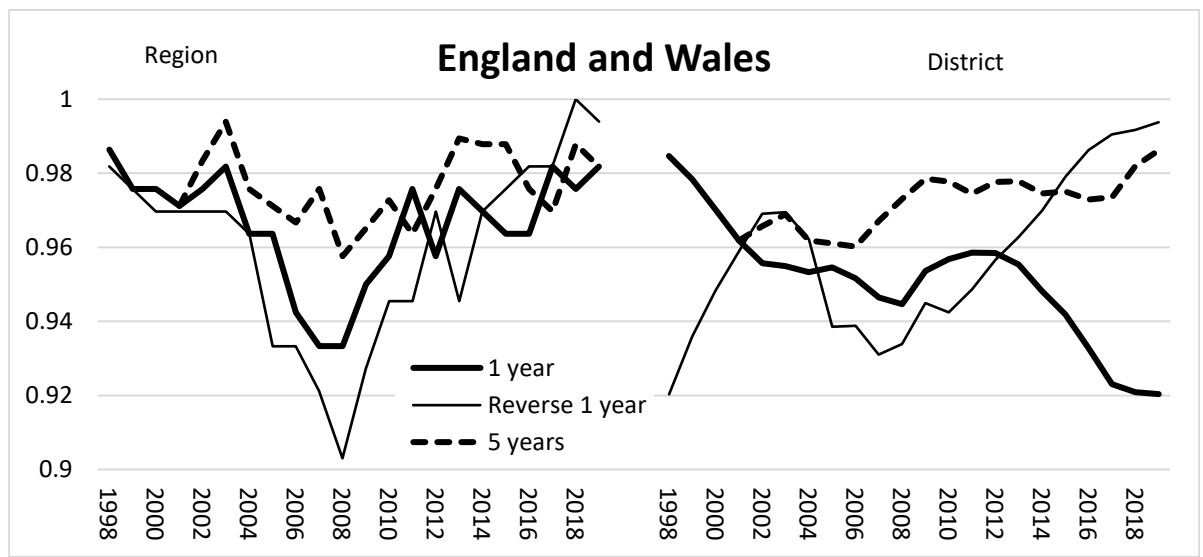

**Figure 2.** Kendall's Concordance.

### 6.2. Beta Divergence

Using three regional groups, each split into City and Other, there was a consideration of equality of median HPER. In rejecting that null and accepting the ordered alternative (Table 1, the J-T statistics for 1997 is 11.34 [0.00]) it confirmed the order that places City-districts in the North as the most affordable group. There was no change in that order by 2019 (14.28 [0.00]), nor the other three years for that matter. Rural were less affordable than urban, and the regional effect trumped the umland effect.

**Table 1.** J-T Growth Test.

|  | North | | Midlands | | South | | J-T | *p*-Value |
|---|---|---|---|---|---|---|---|---|
|  | City | Other | City | Other | City | Other |  |  |
| 1997–2019 [1] |  |  |  |  |  |  | 10.70 | 0.00 |
| 1997 [1] | 0.79 [2] | 0.86 | 0.84 | 0.97 | 1.09 | 1.19 | 11.34 | 0.00 |
| 1997–2004 [1] |  |  |  |  |  |  | 6.21 | 0.00 |
| 2004 | 0.69 | 0.79 | 0.80 | 1.00 | 1.15 | 1.23 | 12.83 | 0.00 |
| 2004–2008 |  |  |  |  |  |  | −3.86 | 0.00 |
| 2008 | 0.79 | 0.89 | 0.79 | 1.00 | 1.18 | 1.25 | 12.59 | 0.00 |
| 2008–2013 |  |  |  |  |  |  | 6.39 | 0.00 |
| 2013 | 0.73 | 0.80 | 0.74 | 0.93 | 1.20 | 1.24 | 12.95 | 0.00 |
| 2013–2019 |  |  |  |  |  |  | 8.17 | 0.00 |
| 2019 | 0.71 | 0.74 | 0.79 | 0.99 | 1.41 | 1.34 | 14.28 | 0.00 |

[1] Restricted pool of districts = 315; [2] HPER for a City district in the North super-region relative to the E&W average.

This six-group order in levels was then related to the growth rates among these groups. Over the full period, the null of common rates of change was rejected (10.7 [0.00] in Table 1); the growth rate-order was found to correspond with the levels-order established above, so

that City-districts in the North grew less affordable at the slowest rate. As the HPER levels-order was found in 2019, the revealed growth-order aligned with it also and indicated the higher-valued districts grew more quickly; hence, a divergent system. This was consistent with a steepening of the HPER distribution seen in Gray (2022). For illustrative purposes, Table 1 reports the harmonic means of the HPER for the terminal period as a ratio of the HPER for the initial. A value less than 1 indicated the growth rate was less than the national average rate.

Following Cook (2012), the full period was broken into sub-periods of 1997–2004, 2004–2008, 2008–2013 and then on to 2019. Sub-dividing the period did not alter the general view that there was an order to the HPERs across E&W. That was not the case for level-growth analysis. For 2004–2008, levels and their growth rate-orders were inversely related (–3.86 [0.00]). In other words, during the house price bubble period, the district HPERs converged, *when ratios were rising*. However, the hierarchy was divergent across 2008–2013 when *all the groups saw a fall in ratios and the average HPER dropped* from 6.897 to 6.736; inconsistent with Cook's (2012) thesis of price convergence-divergence.

*6.3. Paths*

Figure 3 features the 10-time paths with an accompanying map to show the spatial distribution of the various profiles. The district profiles of UDD and DDD were not distributed randomly. Long-term declining cities in the North, such as Sheffield and Liverpool, and the continuous decline of Newcastle were surrounded by rural deterioration. Sparsely populated areas, such as rural Wales, Cornwall and East Yorkshire, had a very weak performance. Although some of the long-term resurgent districts were outside the south east, such as York and Bristol, the continuous growers included much of London, and its commuting towns, such as Brighton and Bedford, that have a rail line running over 160 km directly north with London at its centre.

The most common profile was DUU (62 districts) followed by UDD (57); a quarter having a persistent profile (UUU or DDD). Indeed, these four covered 60% of districts. The least common paths were those that implied little net change (∧ and W1&2), supporting T&M and Glaeser and Gyourko's (2005) in a persistent trend thesis, rather than shock or cycle. This would accord with a path-dependency explanation of persistence. Moreover, the general trends for the cities mentioned above also feature in Gray (2021), with York and Bristol seeing an increase, and Sheffield, Newcastle and Liverpool a decrease in relative populations.

In Table 2, there is a regional and national summary of the 10 profiles. The harmonic means of the raw HPER values for each of the 10 profiles for 2004 and 2019 are reported as per Table 1. In a period of rising HPERs, those that persistently declined, relatively, maintained their ratios, whereas the ratios of persistent risers increased from 7.65 to 11.65, the largest surge. The *average* HPER for E&W rose from 5.85 to 7.85. As such the UUU group's ratio increased from 1.31 times to 1.48 times the mean for E&W. As they started from above average and pulled away, the profile points to divergence. DDU and DDD districts dropped from above to below average, with the latter group falling behind by as much as the UUU group pulled ahead, again signifying divergence but at the bottom end. The other, central profiles had relatively few cases.

It was clear the UUU, UUD and DUU profiles were most commonly linked to London, either directly or being within its commuter zone. Along with the extensive transport infrastructure, London's HMA fits with De Goei et al.'s (2010) super region of the south east. The upward trends were not inconsistent with a productivity divide getting worse from the mid-2000s (Gal and Egeland 2018); an escalator region and self-sorting of the most productive (Martin et al. 2014).

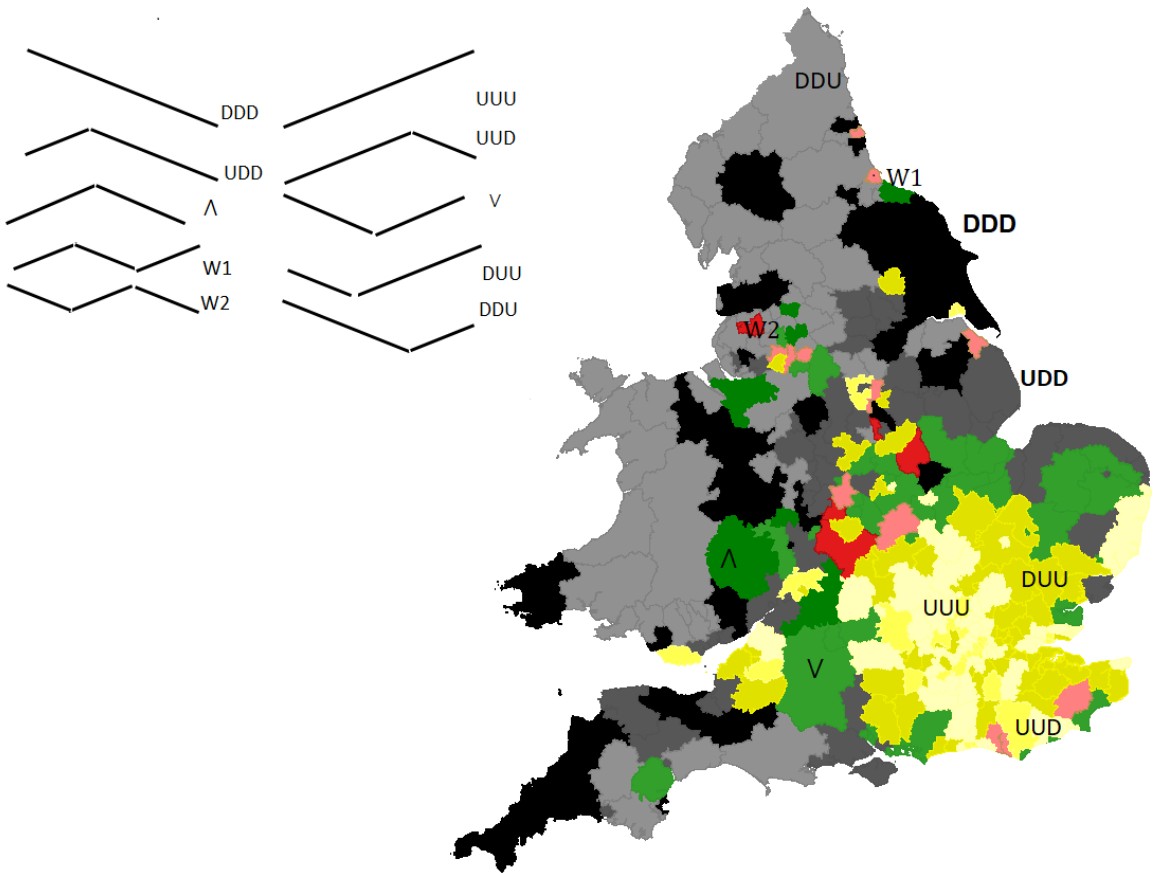

**Figure 3.** The Spatial Distribution of HPER Growth Trajectories. DDD = Continuous decline; ∧ = Medium-term decline; UUD = Recent decline; DUU = Long-term resurgence; UDD = Long-term decline; W1, W2 = Growth set-back; ∨ = Medium-term resurgence; DDU = Recent resurgence; UUU = Continuous growth.

Applying the J-T test in trends, the null that districts, when grouped by profile, have a common HPER was rejected. The alternative was that those with a relatively *high* HPER in 2004 were likely to have an ascending path over 2004–2019 (J-T statistic = 6.507 [0.000] in Table 2). Interestingly, this was also the case when using the 2019 rank order (12.59 [0.000]). With both periods linking ascending trends with high HPER levels, there was a splaying of HPERs rather than convergence, consistent with the findings above with growth rates.

Table 2 also sub-divides the national picture into the three super-regions: North, Midlands and South. Using the initial year, the statistic (J-T = 1.365 [0.172]) points to no difference in medians among districts of the South. By the terminal year the J-T statistic (7.773 [0.000]) indicated a positive relationship between ascending profile and HPER, which was consistent with divergence. The Midlands' results were similar to the South's. The North had relatively low initial HPERs among the relative growers and high among the decliners. The 2004 statistic (−3.33 [0.001]), combined with the insignificant value (−0.458 [0.647]) for 2019, points to convergence over the period with unequal initial but equal terminal medians. Table 2 reports the distribution of the profiles for each super-group subdivided by City and Other. Rural (Other) North was dominated by adverse trends.

The last set of values in Table 2 highlight the region as opposed to regional groups. Divergence was found within the large arc of regions around London. Interestingly, there was no pattern within London to comment on ($p$ = 0.373 and 0.322), which could reflect tight co-movement of prices in the Capital in an integrated market.

**Table 2.** Jonckheere-Terpstra trend tests. Regional Groups: Midlands: EM = East Midlands, WM = West Midlands, South: EE = East of England, LON = London, SE = South East, SW = South West, North: NE = North East, NW = North West, WA = Wales, YH = Yorkshire/Humberside.

| Regional Group | | DDD [1] | DDU | UDD | ∨ | ∧ | W1 | W2 | DUU | UUD | UUU | J-T | *p*-Value [2] |
|---|---|---|---|---|---|---|---|---|---|---|---|---|---|
| North | 2004 [3] | 1.03 | 0.91 | 0.79 | 0.96 | 0.58 | 1.07 | 0.69 | 1.13 | 0.70 | | −3.33 | 0.001 [#] |
| | 2019 [4] | 0.77 | 0.76 | 0.67 | 0.92 | 0.66 | 0.87 | 0.71 | 1.13 | 0.71 | | −0.458 | 0.647 |
| | City [5] | 6 | 7 | 17 | 2 | 3 | 4 | 1 | 2 | 1 | | | |
| | Other [6] | 14 | 1 | 32 | 0 | 1 | 2 | 0 | 0 | 1 | | | |
| Mid-Lands | 2004 | 1.09 | 0.97 | 0.96 | 1.10 | 1.34 | 1.17 | 0.88 | 1.11 | 1.01 | 0.92 | 0.613 | 0.540 |
| | 2019 | 0.86 | 0.80 | 0.83 | 0.99 | 1.03 | 1.08 | 0.89 | 1.04 | 0.91 | 1.01 | 3.437 | 0.001 [~] |
| | City | 4 | 6 | 3 | 3 | 0 | 0 | 2 | 2 | 0 | 1 | | |
| | Other | 6 | 15 | 1 | 13 | 2 | 3 | 2 | 4 | 1 | 2 | | |
| South | 2004 | 1.48 | 1.28 | 1.64 | 1.22 | 1.84 | 1.30 | | 1.28 | 1.56 | 1.34 | 1.365 | 0.172 |
| | 2019 | 1.06 | 1.08 | 1.36 | 1.11 | 1.71 | 1.30 | | 1.32 | 1.77 | 1.53 | 7.730 | 0.000 [~] |
| | City | 0 | 6 | 0 | 9 | 0 | 0 | | 19 | 11 | 23 | | |
| | Other | 5 | 13 | 4 | 12 | 1 | 2 | | 35 | 11 | 23 | | |
| City | 2004 | 0.89 | 0.93 | 0.76 | 1.04 | 0.51 | 0.76 | 1.09 | 1.17 | 1.38 | 1.32 | 7.119 | 0.000 [~] |
| | 2019 | 0.71 | 0.77 | 0.66 | 0.95 | 0.61 | 0.66 | 0.93 | 1.24 | 1.70 | 1.58 | 9.358 | 0.000 [~] |
| | 124 | 10 | 19 | 12 | 14 | 3 | 4 | 3 | 23 | 12 | 24 | | |
| Total | 2004 | 1.10 | 1.07 | 0.83 | 1.15 | 0.79 | 1.15 | 0.81 | 1.26 | 1.39 | 1.31 | 6.507 | 0.000 [~] |
| | 2019 | 0.83 | 0.89 | 0.70 | 1.05 | 0.82 | 1.03 | 0.82 | 1.28 | 1.53 | 1.48 | 12.59 | 0.000 [~] |
| | 338 | 35 | 48 | 57 | 39 | 7 | 11 | 5 | 62 | 25 | 49 | | |
| | | | | | | | | | | | | 2004 [2] | 2019 [2] |
| Region | NE [7] | 4 | 0 | 5 | 0 | 1 | 2 | 0 | 0 | 0 | 0 | 0.013 | 0.936 |
| | NW | 6 | 2 | 21 | 2 | 3 | 3 | 1 | 1 | 0 | 0 | 0.123 | 0.426 |
| | YH | 5 | 4 | 9 | 0 | 0 | 1 | 0 | 1 | 1 | 0 | 0.038 | 0.432 |
| | WA | 5 | 2 | 14 | 0 | 0 | 0 | 0 | 0 | 1 | 0 | 0.362 | 0.478 |
| | WM | 6 | 9 | 3 | 6 | 2 | 1 | 2 | 1 | 0 | 0 | 0.303 | 0.025 |
| | EM | 4 | 12 | 1 | 10 | 0 | 2 | 2 | 5 | 1 | 3 | 0.867 | 0.007 [~] |
| | SW | 5 | 9 | 4 | 3 | 1 | 0 | 0 | 4 | 2 | 1 | 0.334 | 0.064 |
| | EE | 0 | 6 | 0 | 8 | 0 | 0 | 0 | 16 | 3 | 12 | 0.237 | 0.000 [~] |
| | LON | 0 | 0 | 0 | 0 | 0 | 0 | 0 | 9 | 9 | 15 | 0.373 | 0.322 |
| | SE | 0 | 4 | 0 | 10 | 0 | 2 | 0 | 25 | 8 | 18 | 0.063 | 0.000 [~] |

[1] Profiles. See Figure 3; [2] *p*-values of the Jonckheere-Terpstra (J-T) Test; [3] District harmonic mean for 2004 ÷ 5.85; [4] District harmonic mean for 2019 ÷ 7.85; [5] Number of City districts with the corresponding profile; [6] Number of Other districts with the corresponding profile; [7] Number of NE districts with the corresponding profile—see Table 1; [#] Sig at 1% level negative order; [~] Sig at 1% level positive order.

The classifications suggest trends were commonplace. Of the 57 districts with UDD profiles, from Table 2, 49 were found in the North. Of the 62 DUU profiles 54 were found in the South. As the former started from below and the latter from above the mean, combined, just under 30% of districts fit the pattern of convergence to 2008 and divergence from then on. Yet concordance points to a stable hierarchy, implying the existence of rank-preserving changes. There was both inter-super-regional convergence and divergence within a national context of divergence. Although these results appear inconsistent, imagine two cones of HPERs on top of each other: the lower, northern one narrowing with time, with the smaller nodes declining more rapidly than City-districts, and the upper one, expanding with time, being the southern one. Here, the City districts leapfrogged Other-districts.

## 7. Conclusions

This paper features the analysis of house price-earnings ratios at regional and district levels, using ranks. The ratios generally rose from a low in 1997 to the bubble period of 2004–2008 and have not really unwound since. However, subsequently, for the south of

England, there has been some continuation of this increase, whereas for other areas, the picture is one of relative stability.

At its heart are conflicting sets of findings. The first, based on concordance, points to a stable hierarchy, both at the regional and district levels. The second set, based on grouping districts into regions and super-regions, suggest that ascending and descending paths in ratios are the norm.

The descenders in the initial period have lower ratios on average than the ascenders, so the super-regional order is unaltered over time, but the gap grows. Within the super-regions there is adjustment. Envisage two cones of HPERs stacked on top of each other: the lower, northern one, narrowing with time, or convergence; the upper, southern one, expanding with time, highlighting divergence. Steepening of the HPER profile indicates the latter outweighs the former.

The bubble period sees the North catch up with the South. One could argue that this is more a reflection of dominant profiles of long-term resurgence and long-term decline that are spatially distributed in such a way as to imitate a cyclical phenomenon with a phase delay, or ripple. There are cycles in affordability. However, the trend is for a greater divide in housing markets. It is posited that house prices and affordability are reflections of human capital movements, finance and productivity. The escalator region, which benefits from locational sorting of those with the greatest human capital, should be limited by the crowding costs that come with concentration. The disincentive may operate with rents, but ownership may be different. Fielding's (1992) escalator analysis specifically mentions a beneficial HMA as an attractor to those with high human capital. An asset-price model projects that finance will favour advantaged districts with bright futures. By self-sorting across space, there is a concentration of higher long-term earners in a very fluid, extended London market. To the lender, the default risk for these characteristics would be such that they are willing to be flexible on lending criteria, operating on higher multiples. By implication, the concentration of talent facilitates the HPER steepening nationally.

This presents three policy issues. First, how are ownership opportunities offered in the south east to those without high human capital but who are necessary for running the services of a city? If renting is the only option, that will have to be subsidised.

Second, the steepening of the distribution of district HPERs, or the dislocation of London's markets, which implies divergence in spatial affordability, could reflect lenders' perceptions of risk-adjusted returns, not necessarily reckless lending or severe unaffordability. The Bank of England seeking to limit reckless lending with a national metric lending rule may not be apposite for most of Britain. Given the argument above, a lower ratio district could pose the same risk to lenders of non-performing mortgages as one in the south east.

Third, in the war for talent, London is the magnet for university graduates, which creates an over concentration of human capital in one place. If the crowding cost in the capital is not sufficient to deter the immigration of the university crop, a tax on the capital gain from dwelling ownership is a potential solution. Taxing the housing capital gain would, at the very least, bolster tax revenue. Given that self-sorting will occur, university cities, such as Bristol and York, with currently favourable population movements, could be more successful in retaining a greater proportion of their graduates (Swinney and Williams 2016); a step towards a higher-wage/productivity economy away from the core region. Lenders' risk appetites under these circumstances could be lower in general, narrowing the range of HPERs nationally.

**Funding:** There is no external fundings.

**Conflicts of Interest:** The authors declare no conflict of interest.

## Notes

1     https://www.centreforcities.org/the-changing-geography-of-the-uk-economy/ (accessed on 1 July 2020).
2     Table 29 Housing market: simple average house prices, mortgage advances and incomes of borrowers, by new/other dwellings, type of buyer and standard statistical region, from 1969 (previously DCLG table 515) ONS.
3     https://www.centreforcities.org/the-changing-geography-of-the-uk-economy/ (accessed on 1 July 2020).
4     https://www.ncl.ac.uk/media/wwwnclacuk/curds/files/primary-urban-areas.pdf (accessed on 1 July 2020).

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
