# Peer review of "How Have District-Based House Price Earnings Ratios Evolved in England and Wales?"

_jrfm, doi:10.3390/jrfm15080351_

Round 1
Reviewer 1 Report
1. This is about an interesting topic.
2. Motivate why you use HPER rather than DTI.
3. Define HPER using earnings and housing prices rather than a ratio of ratios (it seems that you made a mistake with ratio of ratios, see below). Further, your current definition seems to differ from Gregoriou et al (2014, Urban Studies).
4. In the last paragraph of section 2 you write "dividing the latter by the former". However, this does NOT yield 4.4. Connect to Gregoriou et al. here. Further, just for exposition, a ratio of rations may be difficult for the reader as well.
Author Response
Thanks. To deal with the issues we thought it best to restructure the relevant section. Combined the following is now included:
Researchers examining time series data on HPERs commonly use Nationwide Building Society or Halifax Bank’s house price data. For example, to generate HPERs, Pitros and Arayici (2017) and Gregoriou et al. (2014) use the earnings values of full-time males taken from the ONS. This approach to generating spatial HPERs is much like Demographia’s and the ONS’s.
Although since 2014 the Bank of England regulates the provision of mortgages through debt-to-income ratio (DTI) and the loan-to-value (LTV) rather than HPER directly, DTI is likely to provide similar indications of financial distress as HPER. The ratio reported by UK Finance, the body representing big UK mortgage providers, and regulated by the Bank of England are based on earnings of the borrower (or borrowers where two incomes are used to assess the feasibility of repayment). UK Finance reported values for December 2019 for a first time buyer (FTB) of a DTI of 3.54 and a LTV of 0.77 and 3.3, 0.68 for others. For 1997, the values were 2.22 0.88 & 1.99 0.65. More is lent for each £1 earned and FTBs are expected to find a larger deposit.
Reviewer 2 Report
1. Format of keywords is not uniform. Kindly keep the format unique.
2. Any two keywords should be separated with a colon which is not the case in the first two.
3. There is no uniform spacing in the headings.
4. Figure and table captions are not centralized.
5. Figure captions should be in a uniform format. For example, ‘figure 3’ is not bold.
6. The description with the figures is not in a unique format.
7. Format of table captions is not unique.
8. The paragraphs are not uniformly spaced in section 3.
9. The headings in Table 1 are not in a uniform format. Some are bold and some are not. Kindly keep them uniform.
10. Kindly keep some space between the description under the table or figure and the text in the article.
11. References should be in the same format. They need not be underlined. Kindly correct it.
12. Journal names are missing in some of the references and some are not italicised.
Author Response
Thanks for these pointers
Rather than go through each one we have highlighted changes in red in the resubmission. We trust they are consistent which what you find on your website.
Round 2
Reviewer 1 Report
You write in the paper "DTI is likely to provide similar indications of financial distress as HPER." Could you report the correlation? Comparing an HPER table to DTI results would be even better.
Split the sentence at the end of section 2 into two. One about first time homebuyers and the other one about those who bought before.
Author Response
You write in the paper "DTI is likely to provide similar indications of financial distress as HPER." Could you report the correlation? Comparing an HPER table to DTI results would be even better.
Split the sentence at the end of section 2 into two. One about first time homebuyers and the other one about those who bought before.
I rewrote the para including to correlations as requested:
Although since 2014 the Bank of England regulates the provision of mortgages through debt-to-income ratio (DTI) and the loan-to-value (LTV) rather than HPER directly, DTI is likely to provide similar indications of financial distress as HPER. The ratio reported by UK Finance, the body representing big UK mortgage providers, and regulated by the Bank of England are based on earnings of the borrower (or borrowers where two incomes are used to assess the feasibility of repayment). UK Finance reported values for December 2019 for a first time buyer (FTB) of a DTI of 3.54 and a LTV of 0.77 whilst for 1997, the values were 2.22, 0.88. More is lent per unit income whilst deposits have grown. The long term correlation between the HPER and DTI is around 0.96. For Others, they come to the market, usually with equity in the home to sell. Their borrowing per unit income has risen from 1.99 to 3.3, whilst the LTVs have remained stable (0.65 to 0.68). The correlation between debt and HPER is lower at 0.89.
Reviewer 2 Report
Paper can be accepted now
Author Response
Reviewer 2 is happy
Round 3
Reviewer 1 Report
Why is it likely that another measure yields similar results when you do not even report the correlation between the measures?